# Diversify, Don't Fine-Tune: Scaling Up Visual Recognition Training with Synthetic Images

**Zhuoran Yu**[1*]  **Chenchen Zhu**[2]  **Sean Culatana**[2]  **Raghuraman Krishnamoorthi**[2]
**Fanyi Xiao**[2†]  **Yong Jae Lee**[1†]
**[1]University of Wisconsin-Madison**   **[2] Meta**

**Reviewed on OpenReview:** `https://openreview.net/forum?id=YCt8lsIDwA`

## Abstract

Recent advances in generative deep learning have enabled the creation of high-quality synthetic images in text-to-image generation. While prior research indicates that fine-tuning a pretrained diffusion model on ImageNet and generating synthetic training images can boost an ImageNet classifier's performance, when synthetic images start to outnumber real ones in training, the classifier performance starts to degrade, underscoring the scalability challenge of training with synthetic data. In this paper, we delve into the necessity of generative fine-tuning for achieving recognition performance improvements and investigate the scalability of training with large-scale synthetic images. We find that leveraging off-the-shelf generative models without fine-tuning, while addressing challenges of class name ambiguity, limited prompt diversity, and domain shifts effectively mitigates performance degradation from large-scale synthetic data. Specifically, we leverage large language models (LLMs) and CLIP to resolve class name ambiguity. To diversify images, we propose contextualized diversification (CD) and stylized diversification (SD) methods, also prompted by LLMs. Finally, to mitigate domain shifts, we leverage domain adaptation techniques with auxiliary batch normalization for synthetic images. Our framework consistently boosts recognition model performance with increased synthetic data, even up to 6 times the original ImageNet size. Models trained with our approach demonstrate significant in-domain improvement on ImageNet-val (1.20% to 2.35% across various architectures) and strong out-of-domain generalization on ImageNet-Sketch and -Rendition ($\sim$10% improvement with large vision transformers).

## 1 Introduction

Recent advances in denoising probabilistic diffusion models (Ho et al., 2020; Nichol & Dhariwal, 2021; Ho et al., 2022; Dhariwal & Nichol, 2021; Rombach et al., 2022) have excelled in generating photo-realistic synthetic images in open-vocabulary text-to-image generation (Ramesh et al., 2022; Saharia et al., 2022; Rombach et al., 2022; Nichol et al., 2021; Yu et al., 2023). These capabilities stem from large-scale image-text datasets like LAION (Schuhmann et al., 2021; 2022) and multimodal models like CLIP (Radford et al., 2021). Ongoing research explores the potential of using synthetic images from such models to improve recognition models (Azizi et al., 2023; He et al., 2022; Sariyildiz et al., 2023), mostly in scenarios with low-shot real training images. Critically, whether such synthetic images can improve a recognition model's performance when real images at the ImageNet-scale are already available remains an open research question.

Recent work (Azizi et al., 2023) tries to answer this question by delving into the intricate process of fine-tuning the open-vocabulary Imagen model (Saharia et al., 2022) on ImageNet (Deng et al., 2009) and using the finetuned model to generate synthetic images conditioned on ImageNet class labels. However, designing

---

*work partially done during internship at Meta.
†equal advising.

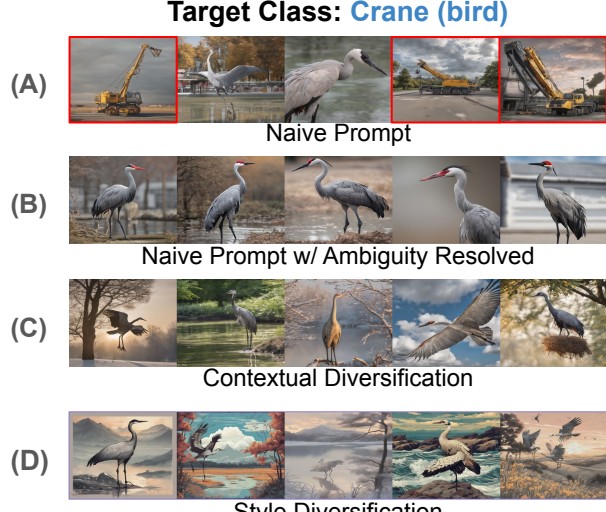

Figure 1: **Prompt augmentation for unambiguous and diversified synthetic data generation**. We improve synthetic data generation by augmenting prompts from two perspectives: 1) we resolve the ambiguity for class names to avoid generating images with incorrect semantics for the target class (*e.g.*, row A *vs.* B), and 2) we diversify the prompts used to generate synthetic images both in terms of their contexts (row C) and styles (row D). We automatically achieve both augmentations with LLMs.

the fine-tuning strategy adds an extra layer of complexity and requires repetitive work when applying to diverse datasets. More importantly, Azizi et al. (2023) demonstrates that exceeding the ratio of synthetic to real images can lead to significant performance degradation. For instance, training with 4.8 million synthetic images alongside default ImageNet training data yields lower accuracy compared to training solely with real images.

We argue that the scalability failure of synthetic images stems from the generative fine-tuning process. Fine-tuning pretrained diffusion models on the ImageNet training set aligns the model and the generated synthetic images with the distribution of ImageNet training data. Consequently, training with an excess of synthetic images from fine-tuned models can lead to notable overfitting to the ImageNet training set, resulting in performance degradation.

To validate this argument, instead of fine-tuning a pretrained open-vocabulary diffusion model, we leverage a frozen off-the-shelf one to generate additional training images to the ImageNet training data. One typical approach to achieving this would be to employ the naive prompt format *a photo of a {class name}* and train the recognition model with a combination of real and synthetic images. However, several notable challenges arise from this simple approach from both the data generation and model training aspects, which lead to sub-optimal recognition performance, as discussed next.

**Synthetic Data Generation.** Datasets such as ImageNet often contain class names that have multiple meanings. For instance, *crane* can denote both a construction machine and a bird species. These divergent interpretations can lead to unintended semantic misalignment between synthetic images and class semantics such as generating machine images for a class of bird cranes. Furthermore, adhering to a fixed prompt format can lead to redundancy in the generated images, where most of them exhibit similar visual characteristics. The limited diversity in synthetic images exacerbates the risk of the recognition model overfitting to such images, thereby leading to challenges when scaling up training with synthetic data.

We address these challenges by redesigning the data generation pipeline to solve the label ambiguity and generate diversified images, with the assistance of large language models (LLMs). While existing approaches leverage dataset-specific hierarchical information (Sariyildiz et al., 2023) or manually process class names (Radford et al., 2021) to resolve the ambiguity in class names, we propose a general solution that requires minimal human effort. Specifically, we query LLMs to extract multiple meanings of each class name and select the meaning with the highest CLIP (Radford et al., 2021) image-text similarity associated with

the actual real images of the class. The selected meaning is used as an additional explanation of the class name in the synthetic data generation process.

Next, we generate diversified synthetic images by formulating generation prompts using the knowledge from LLMs. We introduce two distinct methods for diversification: *contextualized diversification (CD)* and *stylized diversification (SD)*, each based on different principles. CD aims to generate photo-realistic synthetic images featuring class *c* with diversity introduced through contextual foreground and background objects. In contrast, SD emphasizes the generation of images with varying styles. This whole process of obtaining generation prompts is realized by prompting LLMs, minimizing the need for human labor.

Compared to finetuning diffusion models on the target dataset (Azizi et al., 2023), our approach is able to generate synthetic images featuring the same class but containing contextual information beyond the original dataset, giving it potential to generalize better on out-of-distribution datasets (as we show in Section 4).

**Domain Shifts in Recognition Model Training.** Another inherent challenge arises from the domain shifts between real and synthetic images. While recent diffusion models produce synthetic images of higher quality, they still bear a different distribution as manifested by their separable features within the feature space (Ojha et al., 2023). When synthetic images start to dominate the real images in training, the domain shifts can lead to suboptimal performance if not properly handled.

We view real and synthetic images as being from two separate domains, and propose to mitigate the domain shifts between them by drawing inspiration from domain adaptation literature. First, we introduce auxiliary batch normalization layers (BN) (Chang et al., 2019; Seo et al., 2020; Li et al., 2018) for recognition models that employ BN to process synthetic images. We further adjust domain-level sampling weights ensuring each batch containing a roughly equal number of real and synthetic images during training, avoiding overfitting to synthetic images.

**Main Contributions.** Through our refined design of the synthetic data generation pipeline and mitigation of domain gaps, our framework yields significant performance improvements over using fine-tuned generative models (Azizi et al., 2023) on ImageNet classification. For example, with ResNet-50, our method achieves +2.53% accuracy improvement over the baseline trained with real images only, and +1.10% compared to training with real and synthetic images from finetuned Imagen (Azizi et al., 2023).

Crucially, our simpler framework (*i.e.*, without generative fine-tuning) consistently enhances results, even when scaling up unique synthetic data to 6x (balanced sampling used in each training batch) of the original training dataset size and further increasing synthetic data only results in minimal performance difference. This is in stark contrast to prior generative fine-tuning work (Azizi et al., 2023), which observed that recognition performance degrades as the synthetic images outnumber real ones in training. This underscores the untapped potential of synthetic data at a larger scale. Additionally, models trained with synthetic images from our framework also show more robust performance for out-of-domain generalization. Specifically, a ResNet-50 trained with ImageNet real images and synthetic images from our framework achieves +2.89%, +7.55%, and +5.77% absolute accuracy improvement on ImageNet-V2 (Recht et al., 2019), ImageNet-Sketch (Wang et al., 2019), and ImageNet-Rendition (Hendrycks et al., 2021), respectively; with vision transformers, the improvements on ImageNet-Sketch and ImageNet-Rendition can even exceed +10%.

## 2 Related Work

**Diffusion Models for Image Generation**. Diffusion models (Sohl-Dickstein et al., 2015) have been widely applied in image generation (Kingma et al., 2021; Nichol & Dhariwal, 2021; Ho et al., 2022; Dhariwal & Nichol, 2021; Rombach et al., 2022). With the introduction of large-scale image-text datasets (Schuhmann et al., 2021; 2022) and image-text foundation models like CLIP (Nichol & Dhariwal, 2021), state-of-the-art diffusion models (Ramesh et al., 2022; Saharia et al., 2022; Rombach et al., 2022; Nichol et al., 2021; Yu et al., 2023) can perform text-to-image generation in an open-vocabulary manner. For example, the latent diffusion model (LDM) (Rombach et al., 2022) conducts diffusion processes in the latent space; Imagen (Saharia et al., 2022), on the other hand, directly runs diffusion steps at the pixel level. Recent works extend these models for better control in image generation (Zhang et al., 2023; Bar-Tal et al., 2023; Li et al., 2023b; Huang et al., 2023; Brooks et al., 2023). Our focus is not to improve state-of-the-art diffusion models; instead, we propose

a new framework that leverages such models to generate synthetic data to improve a recognition model's performance on large-scale datasets.

**Improving Recognition Models with Synthetic Data.** We position our work within the line of work that studies how to improve visual recognition models with synthetic data from 3D-rendering (Hesse et al., 2023; Fu et al., 2023; Yang et al., 2023; Zheng et al., 2023), simulation environments (Richter et al., 2016; Dosovitskiy et al., 2017; Gan et al., 2020; de Melo et al., 2022), or generative models (Li et al., 2021; 2022; Gowal et al., 2021; He et al., 2022; Lin et al., 2023; Ali-Gombe et al., 2018). Early work (Li et al., 2022; 2021; Ali-Gombe et al., 2018) typically trained a generative adversarial network (Goodfellow et al., 2014; Karras et al., 2019; 2020; Brock et al., 2018) on the target dataset to generate synthetic images. Recent work explores the potential of synthetic images from diffusion models to improve visual recognition (He et al., 2022; Sariyildiz et al., 2023; Lin et al., 2023; Wu et al., 2024). For example, (He et al., 2022; Li et al., 2023a) shows that CLIP classification on a target dataset can be improved with synthetic data from diffusion models; Sariyildiz et al. (Sariyildiz et al., 2023) uses only synthetic data to train image classification models from scratch; Lin et al. (Lin et al., 2023) extends this study to object detection by pasting object-centric synthetic images onto training data under the few-shot scenario. This line of research primarily addresses scenarios with limited or no real training data, whereas our work concentrates on investigating the scalability of model training with synthetic data when an ImageNet-scale real training dataset is already accessible. Most related to our work, Azizi et al. (Azizi et al., 2023) recently showed that synthetic data from Imagen (Saharia et al., 2022) finetuned on ImageNet can improve recognition accuracy on ImageNet but it suffers from performance degradation when the synthetic images start to overwhelm the real images. In contrast, our work offers a streamlined framework without generative fine-tuning. It effectively mitigates performance degradation when scaling up synthetic images during training, and showcases significantly improved performance in both in-domain and out-of-domain evaluations.

## 3 Approach

In this section, we introduce our framework from two main aspects: synthetic data generation and using that data for recognition model training. We leverage a latest variant of an off-the-shelf LDM (Rombach et al., 2022) without any finetuning to generate our synthetic images, and train our recognition models using both real training images and generated synthetic images with a carefully designed training recipe.

### 3.1 Label Ambiguity Resolution

Real-world datasets like ImageNet (Deng et al., 2009), often present a challenge where class labels can carry multiple divergent meanings. For instance, the term *crane* could refer to a type of construction machine or a species of bird. These dual interpretations lead to generations with significantly divergent semantics if not addressed properly: construction machine images are generated when the intended class is bird. While existing solutions tackle this problem by leveraging dataset-specific hierarchical information (Sariyildiz et al., 2023), finetuning (Azizi et al., 2023), or manual processing (Radford et al., 2021), we propose a more comprehensive and, importantly, manual effort-free approach.

Our approach is centered around each class name $c$. Rather than manually curating a specific hierarchy for class names, we turn to LLMs for assistance. Given class $c$ with $N$ real training images, we query an LLM to extract $K$ possible meaning phrases $\{P_i^c\}_{i=1}^K$ associated with $c$ and compute their text embeddings $\{\phi(P_i^c)\}_{i=1}^K$ using the CLIP (Radford et al., 2021) text encoder $\phi$. We also extract image embeddings $\{\theta(I_j^c)\}_{j=1}^N$ for the real training images $\{I_j^c\}_{j=1}^N$ using the CLIP visual encoder $\theta$. We then compute the image-text similarities and select the meaning phrase with highest similarity $\hat{P}^c$:

$$\hat{P}^c = \arg\max_{P_i^c} \sum_{j=1}^N \theta(I_j^c) \cdot \phi(P_i^c) \tag{1}$$

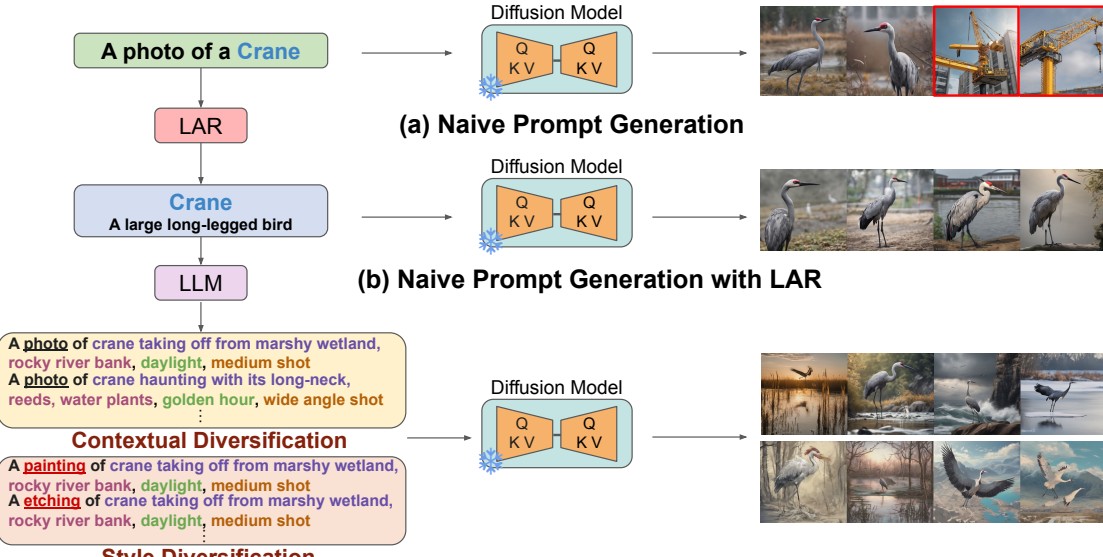

Figure 2: **Our synthetic data generation pipeline.** (a) Generating synthetic data with naive prompts can lead to incorrect semantics for classes with ambiguous names (*e.g.*, the bird *vs.* the machine for "crane"). (b) Our Label Ambiguity Resolution (LAR) procedure resolves ambiguity in labels while preserving similar semantics in the generated images. (c) Our diversification procedure includes contextual diversification (CD) and style diversification (SD) by prompting an LLM to produce contextualized descriptions of images featuring class $c$ ("crane" in this example) that combines different aspects (indicated by different colors in the figure): **foreground objects**, **background objects**, **lighting condition**, **camera angle**, and different **styles**.

This process is demonstrated in Figure 3 and it helps pinpoint the most similar meaning $\hat{P}^c$ which is then used as an additional textual description alongside class name $c$ during synthetic data generation (e.g., "A large long-legged bird" for 'Crane'). By leveraging LLMs and CLIP, we minimize manual effort and ensure that the synthetic data accurately represents the intended semantics even in situations where the class name has multiple meanings; see Figure 2 (b).

## 3.2 Diversifying Synthetic Images

The process of diversifying synthetic images goes beyond a mere reliance on class names ($c$) and their associated meanings ($\hat{P}^c$). Our approach introduces two distinct methods for diversification: contextualized diversification (CD) and stylized diversification (SD), each founded on different principles and mechanisms.

**Contextual Diversification (CD)** is dedicated to producing photo-realistic synthetic images for each class $c$ while maximizing diversity in their contexts. In CD, diversity is achieved through the incorporation of contextual foreground and background objects, which enrich the visual content of the synthetic images. Specifically, we prompt an LLM to produce contextualized descriptions of images featuring class $c$ in a specific way that combines foreground objects, background objects, lighting condition, and camera angle. An example is shown as follows:

```
Imagine there is a photo of {class c}. What foreground and
background objects can show up together with it? Describe
the photo in the following four aspects:
— Foreground
— Background
— Lighting Condition
— Camera Angle
```

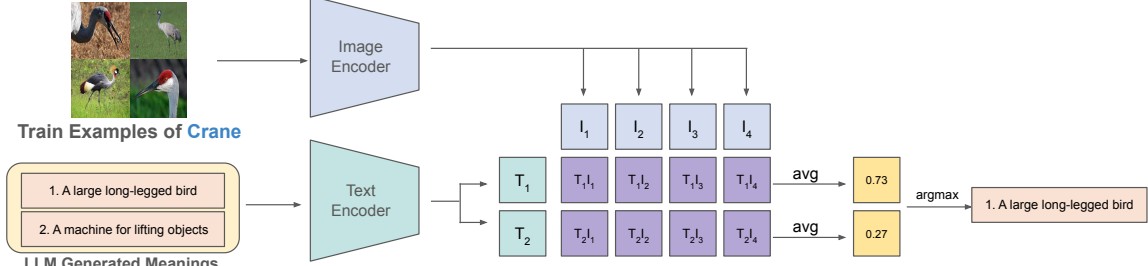

Figure 3: **Label Ambiguity Resolution.** Given multiple meanings of class name $c$ (crane in this example), we leverage CLIP to compute average similarity between each meaning and training examples of class $c$. Then, we use averaged image-text similarity as selection metric to select correct meaning of the $c$. In this example, *a large long-legged bird* is selected and used as additional context in the rest of our pipeline.

Prompting an LLM in this way results in better responses than simply asking for a caption of a specific object class, as the LLM may return a generic description of the class name without much context. To maximize diversity in synthetic images, we ask the LLM for 20 descriptions per prompt and explicitly instruct it to "be creative and avoid repetition". We also use a high temperature of 0.75 when prompting the LLM which helps to provide diverse and creative descriptions. We collect 600 total responses from the LLM for each class and form the generation prompt by joining the descriptions from each aspect with a comma and the prefix *a photograph of* to promote photo-realism. When generating synthetic images, we use a guidance scale of 2.0. Since guidance scale controls how strict the diffusion model follows the text prompts, setting a lower guidance scale further encourages generation diversity, and our ablation study indeed shows that such a choice results in the best performance (7c).

The inclusion of these contextual elements not only provides a more diverse and intricate descriptions of the object class, but more importantly, it helps overcome any bias in the original dataset by incorporating knowledge from the LLM in generation. Compared to finetuning diffusion models on the target dataset (Azizi et al., 2023), our approach can generate synthetic images featuring the same class but containing contextual information beyond the original dataset, giving it potential to generalize better on out-of-distribution datasets (as we show in Section 4).

**Style diversification (SD)** is built on top of contextual diversification with an emphasis on generating synthetic images that exhibit a wide range of artistic styles. This approach goes beyond mere realism and delves into the realm of creative expression, allowing for a broader spectrum of synthetic images that cater to diverse aesthetic preferences. Specifically, we query an LLM for 60 different art styles (the list is provided in the Appendix) and replace the keyword "photograph" in the prompts from CD with each art style (*e.g.*, "a photograph of" → "a painting of") to form the new generation prompts. We generate the same number of synthetic images using prompts from CD and SD; see Fig. 2 (c).

The whole process of diversification is summarized in Figure 2. Since both CD and SD are realized with the help of an LLM, it not only automates the process to save human labor, but also ensures that the diversification of synthetic images is achieved in a consistent manner and at scale.

### 3.3 Mitigating Domain Shifts in Recognition Model Training

While recent diffusion models excel in producing high-quality, photo-realistic synthetic images, the distribution gap remains between these synthetic images and their real counterparts. This divergence is most notably observed in the nearly separable features that characterize these images within the feature space (Ojha et al., 2023).

As the scale of synthetic images grows, the risk of the recognition model overfitting on synthetic images also grows. Such overfitting can significantly undermine the recognition model performance, especially when synthetic images start to dominate the training data. As shown in prior work (Azizi et al., 2023), the overall accuracy starts to drop when synthetic data outnumbers real ones and it leads to -2.69% accuracy drop when synthetic data becomes 9x of real data. To circumvent this issue, We view real and synthetic images as from two separate domains inspired by domain adaptation literature. This realization allows us

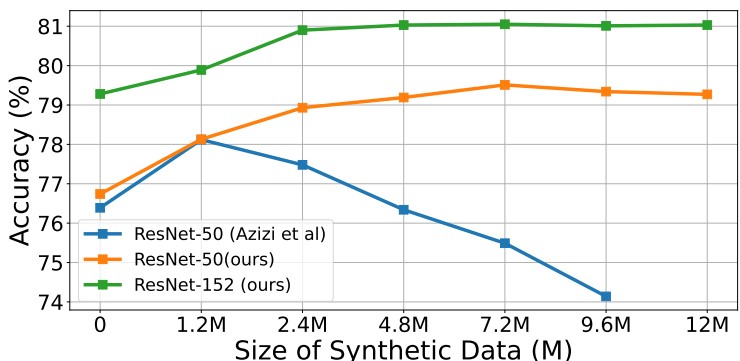

Figure 4: **Top-1 ImageNet Classification Accuracy vs Synthetic Data Size**. In contrast to findings in previous work (Azizi et al., 2023) (blue line), our method is able to scale up recognition training, with consistent accuracy improvement as the number of unique synthetic samples increases.

to introduce auxiliary BN layers (Chang et al., 2019; Seo et al., 2020; Li et al., 2018) into the recognition model to separately process synthetic images from real ones. These additional BN layers play a pivotal role in bridging the domain gap as they prevent the batch statistics of synthetic data from disturbing the running means and variances for real images.

In addition, despite the potentially larger scale of synthetic images relative to real images, we enforce equal sampling weights for both the real and synthetic images during training so that each training batch roughly contains the same number of real and synthetic images. This avoids synthetic data from dominating each training batch and thus leading to domain overfitting. The loss we use to train the recognition model is: $\mathcal{L} = -\sum_{i=1}^{N_{\text{real}}} y_i \log(f(x_i)) - \lambda \cdot \sum_{j=1}^{N_{\text{synthetic}}} y_j \log(f'(x_j))$ where $f$ and $f'$ denotes the model weights for real and synthetic images (i.e., they share weights except for BN layers) and $\lambda$ is a tunable hyper-parameter which enables explicit control of the contribution of synthetic data in training.

## 4 Experiments

**Datasets and Evaluation.** We train our recognition model using real images from ImageNet (Deng et al., 2009) and synthetic data generated with our pipeline. Each model is evaluated on ImageNet-val as the in-distribution evaluation. We further consider three ImageNet variations: ImageNet-V2 (Recht et al., 2019), ImageNet-Sketch (Wang et al., 2019), ImageNet-Rendition (Hendrycks et al., 2021) to evaluate out-of-distribution generalization capability. ImageNet-V2 is a reproduced version of ImageNet classes with distribution shifts; ImageNet-Sketch and ImageNet-Rendition contains sketch and rendition versions of ImageNet classes, respectively.

**Implementation Details.** Following prior work (Azizi et al., 2023), we evaluate our approach using different CNNs (ResNet-{50, 101, 152}) and vision transformer (DeiT-{S, B, L} (Touvron et al., 2021)) architectures. We use *gpt-3.5-turbo* (OpenAI, 2022) from OpenAI as our LLM for contextual diversification and style diversification. For each class, we prompt the LLM to generate 600 contextual-diversified prompts and 60 style keywords to form style-diversified prompts in synthetic data generation. We use the latest variation of LDM (Rombach et al., 2022) to generate our synthetic data. All synthetic images are generated with size of 1024x1024 and downsampled to 256x256 for efficient storage and higher throughput in data loading. To fairly evaluate the impact of generative finetuning when using generated synthetic data in training, we closely follow the training setup from prior work (Azizi et al., 2023), which augments ImageNet training with synthetic images from fine-tuned Imagen (Saharia et al., 2022). Due to the unavailability of Imagen and the finetuned model on ImageNet, we believe this is the best practice for fair comparison. Specifically, we train ResNet models for 130 epochs and all vision transformer variants for 300 epochs. We set the synthetic loss weight $\lambda$ to 0.6 for all experiments, which is selected based on in-domain evaluation of ResNet-50 alone. Full details of our hyper-parameters can be found in Appendix.

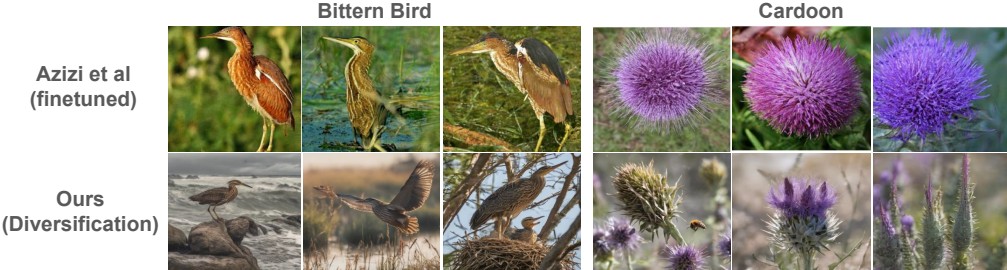

Figure 5: **Qualitative comparison between our diversified synthetic images and synthetic images from (Azizi et al., 2023)**. Since Azizi et al. (2023) does not share their finetuned model and synthetic data, we use the provided qualitative results in their manuscript for analysis. Synthetic images from Azizi et al. (2023) lacks of diversity in both foreground and background whereas our diversified images shows diverse foreground objects in different postures and camera angles and different background environment. The diverse semantic information avoids recognition model overfitting to specific details of synthetic data and prevents performance degradation when scaling up synthetic images.

## 4.1 Scaling-up Training with Synthetic Data

We first demonstrate how recognition performance changes as we scale up the size of synthetic data from 1.2M to 12M. Since ImageNet has roughly 1.2M real training images, our selected range of synthetic data size increases from 1x ImageNet scale to 10x ImageNet scale. We use the same amount of synthetic images from contextual diversification and style diversification for our models.

Figure 4 shows the results. In prior work (Azizi et al., 2023), ResNet-50 suffers performance degradation as the size of synthetic data increases: the accuracy becomes even worse than the baseline trained with only real images when synthetic data exceeds 4x ImageNet scale (4.8M). In stark contrast, models trained with our approach demonstrate consistent improvement up to 6x unique synthetic images at ImageNet scale (while balanced sampling is used). Also unlike (Azizi et al., 2023), which observed significant performance degradation when further scaling up the synthetic images, our approach produces consistent model quality even beyond the point of saturation.

This distinction arises from the core contribution of this paper – it's important to avoid generative fine-tuning, as done in Azizi et al. (2023), to mitigate overfitting when training on synthetic images. Since Azizi et al. (2023) did not release their finetuned model and synthetic training data, we can't directly reproduce and ablate their approach. However, we believe scalability challenges arise from generative finetuning, which causes overfitting to the ImageNet training distribution, resulting in less diverse synthetic images. As the evidence, we provide qualitative comparisons between our diversified synthetic images and synthetic images from Azizi et al. (2023) for better understanding of the performance degradation when scaling up synthetic images in training. As can be seen in Figure 5, synthetic images from Azizi et al. (2023) lacks foreground and background diversity even with a low guidance scale (generally, a low guidance encourages diversity of synthetic images) and training with a large amount of such images risks overfitting to repetitive details (e.g, layout, fore-/background), resulting in poor generalization. In contrast, our diversification methods promoted by CD and SD produce more diversified synthetic images in foreground and background and thus avoids overfitting to synthetic images and encourages better generalization.

## 4.2 Main Results

We next present the main results of our approach with different recognition architectures and classification tasks. In Table 1, we report results of training with ImageNet real images and our diversified synthetic images for each architecture.

When evaluating on ImageNet-val, our approach demonstrates significant and consistent improvement over both the baseline model trained with only real data, and the prior state-of-the-art that uses an ImageNet finetuned diffusion model to generate synthetic data (Azizi et al., 2023). For example, our approach achieves +2.53% and +1.10% absolute accuracy improvement over the real image baseline and prior approach (Azizi et al., 2023) with ResNet-50, respectively. Similar improvements are also observed with larger CNN models

| | ImageNet-val | | | | ImageNet-V2 | | | ImageNet-S | | | ImageNet-R | | |
|---|---|---|---|---|---|---|---|---|---|---|---|---|---|
| | Real Only | Azizi et al Azizi et al. (2023) | Ours | Δ | Real Only | Ours | Δ | Real Only | Ours | Δ | Real Only | Ours | Δ |
| ResNet-50 | 76.74 | 78.17 | **79.27** | +2.53 | 72.20 | **75.09** | +2.89 | 25.01 | **32.56** | +7.55 | 24.55 | **30.32** | +5.77 |
| ResNet-101 | 79.01 | 79.74 | **80.76** | +1.75 | 74.75 | **76.89** | +2.14 | 27.92 | **33.32** | +5.40 | 27.08 | **31.22** | +4.14 |
| ResNet-152 | 79.28 | 80.15 | **81.05** | +1.77 | 75.11 | **77.32** | +2.21 | 29.98 | **33.62** | +3.64 | 28.29 | **32.45** | +4.16 |
| DeiT-S | 78.97 | 80.49 | **81.32** | +2.35 | 74.66 | **77.61** | +2.95 | 28.49 | **42.29** | +13.80 | 27.93 | **39.68** | +11.75 |
| DeiT-B | 81.79 | 82.84 | **82.99** | +1.20 | 77.35 | **79.16** | +1.81 | 36.31 | **47.65** | +11.34 | 34.47 | **44.14** | +9.67 |
| DeiT-L | 82.22 | 83.05 | **83.53** | +1.31 | 78.82 | **79.45** | +0.63 | 39.90 | **50.01** | +10.11 | 37.21 | **46.48** | +9.47 |

Table 1: **Top-1 Accuracy on ImageNet and out-of-domain variations**. We train each model with real ImageNet training images and synthetic images generated with our pipeline. In-domain evaluation is conducted on ImageNet-val (underlined) and out-of-domain evaluations are conducted on ImageNet-V2, ImageNet-Sketch and ImageNet-Rendition. Specifically, we apply the same model trained with ImageNet real and our synthetic images on these out-of-domain evaluation datasets without further finetuning for evaluation. Δ denotes the margin of improvement of our approach over baseline models trained with real images only.

| | Training Data | N=100 | N=200 | N=500 |
|---|---|---|---|---|
| ResNet50 | Real Only | 47.83 | 58.76 | 68.04 |
| ResNet50 | Real + 2x Syn | **60.95** | **65.19** | **73.11** |
| ResNet101 | Real Only | 48.09 | 59.64 | 69.25 |
| ResNet101 | Real + 2x Syn | **62.01** | **67.39** | **74.87** |
| ResNet152 | Real Only | 48.05 | 60.33 | 70.91 |
| ResNet152 | Real + 2x Syn | **63.53** | **69.05** | **75.63** |

Table 2: **Low-data Regime Evaluation.** Our models demonstrate consistent improvement over models trained with only real images. Here, 2x means 2x of the ImageNet data scale (i.e, 2.4M)

like ResNet-101 and ResNet-152, where our approach achieves clear improvements (1.02% and 0.90% respectively) over prior work (Azizi et al., 2023). For vision transformer DeiT-{B, L}, which already demonstrate strong performance on ImageNet, our method can still further improve their performance. These results validate the benefit of our framework in leveraging synthetic data at a large scale across model architectures.

Furthermore, our method shows strong performance on out-of-distribution generalization. When evaluated on ImageNet-V2, ImageNet-Sketch, and ImageNet-Rendition, our method achieves +3% to +7% accuracy improvements over the real-only CNN baselines. With vision transformers, the improvement is even more significant: Deit-B trained with our diversified synthetic data achieves 11.34% and 9.67% accuracy improvement on ImageNet-Sketch and ImageNet-Rendition over the real baseline, respectively.

These results demonstrate the efficacy of diversified synthetic data to improve both in-domain and out-of-domain generalization.

## 4.3 Results for Low-data and Long-tail Settings

While the primary focus of this work is to investigate the scalability of training with ImageNet-scale real training data, we also demonstrate the effectiveness of our approach in low-data and long-tail scenarios in this section.

**Evaluation on Low-data Regime**. Here we demonstrate the effectiveness of our method in settings where only limited real training images are available. Specifically, we sample {100, 200, 500} real training images per class to form the real training data and use 2x additional synthetic data of ImageNet scale (2.4M) from our pipeline. We train each recognition model for 100 epochs as models usually converge faster under the low-data regime.

As shown in Table 2, our improvements on ImageNet transfer well to the low-data regime. With 100 real images per class, additional 2x synthetic data usually brings +10% accuracy improvement. With 500 real images per class, our model achieves 74.87% and 75.63% for ResNet-101 and ResNet-152, which largely closes the gap to training with full ImageNet images (79.01% for ResNet-101 and 79.28% for ResNet-152).

**Evaluation under long-tail distribution.** We also show our pipeline enables strong improvements when the real images follow a long-tailed distribution. For comprehensive evaluation under different imbalance

| | Training Data | $\gamma = 50$ | | | $\gamma = 100$ | | | $\gamma = 200$ | | |
|---|---|---|---|---|---|---|---|---|---|---|
| | | Many-shot | Medium-shot | Few-shot | Many-shot | Medium-shot | Few-shot | Many-shot | Medium-shot | Few-shot |
| ResNet-50 | Real | 78.63 | 69.08 | 45.69 | 79.02 | 68.12 | 39.35 | 78.99 | 66.58 | 33.31 |
| ResNet-50 | Real + 2x Syn | **82.19** | **72.43** | **51.18** | **81.36** | **71.35** | **45.56** | **81.61** | **69.78** | **39.74** |
| ResNet-152 | Real | 80.60 | 72.09 | 49.19 | 80.77 | 70.25 | 42.81 | 80.95 | 68.72 | 36.26 |
| ResNet-152 | Real + 2x Syn | **82.96** | **75.51** | **54.50** | **83.28** | **74.48** | **49.12** | **83.29** | **72.88** | **43.52** |

Table 3: **Evaluation of Long-tail Distribution for Real Images**. We evaluate each model with an imbalance ratio $\gamma$ in {50, 100, 200} by subsampling ImageNet real training images and using 2x additional synthetic data of original ImageNet scale (2.4M) from our pipeline to train the models. Our approach shows consistent performance improvements, especially on few-shot classes.

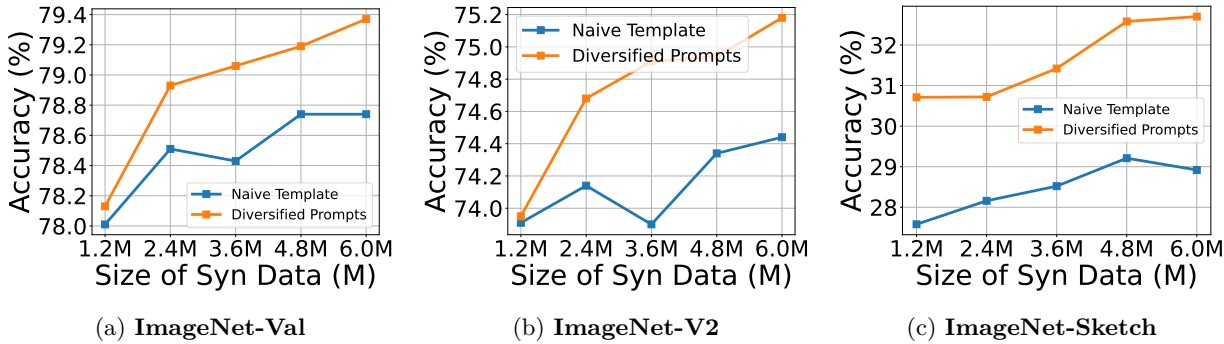

(a) **ImageNet-Val**    (b) **ImageNet-V2**    (c) **ImageNet-Sketch**

Figure 6: **Naive *vs*. Diversified Prompts**. As the synthetic images increase from 1.2M to 6M, we observe a consistent improvement from diversification of synthetic data, and such improvement becomes more distinct as the scale of synthetic data increases. This trend is consistent across different evaluation sets.

ratios, we randomly sample a long-tailed distribution with imbalance ratio $\gamma$ in {50, 100, 200} from the original ImageNet training data, following an exponential imbalance function (Cui et al., 2019). Specifically, the number of real images for class $k$ is computed as $n_k = n_1 \cdot \gamma^{-(k-1)/(K-1)}$, where $K$ is the total number of classes. $n_1$ is the number of labels for the most frequent class and is set to 1300, which is the max number of images per class on ImageNet. Evaluation is conducted on ImageNet-val, which is roughly balanced. We report the accuracy for many-shot classes, medium-shot classes, and few-shot classes. Results are shown in Table 3 and models trained with our synthetic data consistently outperform the real-only models over many-shot, medium-shot and few-shot classes. For few-shot classes, our approach can even improve by 5% to 7% accuracy, which demonstrates the efficacy of diversified synthetic data to tackle long-tail problems.

## 4.4 Ablation Studies

We next present ablation studies. Unless stated otherwise, experiments in this section utilize 2.4M synthetic images generated with both CD and SD.

**Importance of Diversifying Synthetic Images.** Figure 6 compares the performance of a ResNet-50 model on ImageNet-val, -V2, and -Sketch when trained with different amounts of synthetic data generated with naive prompts *A photo of {c}* vs. our diversified prompts described in Section 3.2. We apply separate BN to train both models for fair comparison. As we increase the size of synthetic images from 1.2M to 6M, we see a consistent improvement from diversified synthetic images on these datasets. The improvement is more significant as the scale of synthetic data increases, which demonstrates the critical role that diversifying synthetic images plays in our framework.

**Importance of Separate BN in CNNs.** Using separate batch norm layers (BN) for real and synthetic images is a critical design in our framework and we demonstrate its importance in Table 4. Using separate BN significantly and consistently improves performance for different ResNet architectures, which validates the importance of this design.

**Sensitivity to Synthetic Loss Weights.** We examine the sensitivity of the tunable hyper-parameter $\lambda$, which controls the overall contribution of synthetic data in training. As shown in Figure 7a, the accuracy of

|  | ResNet-50 | ResNet-101 | ResNet-152 |
|---|---|---|---|
| Vanilla BN | 78.32 | 79.92 | 80.43 |
| Separate BN | **78.93** | **80.34** | **81.05** |

Table 4: **Vanilla *vs*. Separate BN**. Results are generated with 2.4M synthetic images with diversification. Separate BN is demonstrated to be effective for training on real+synthetic data, across various model architectures.

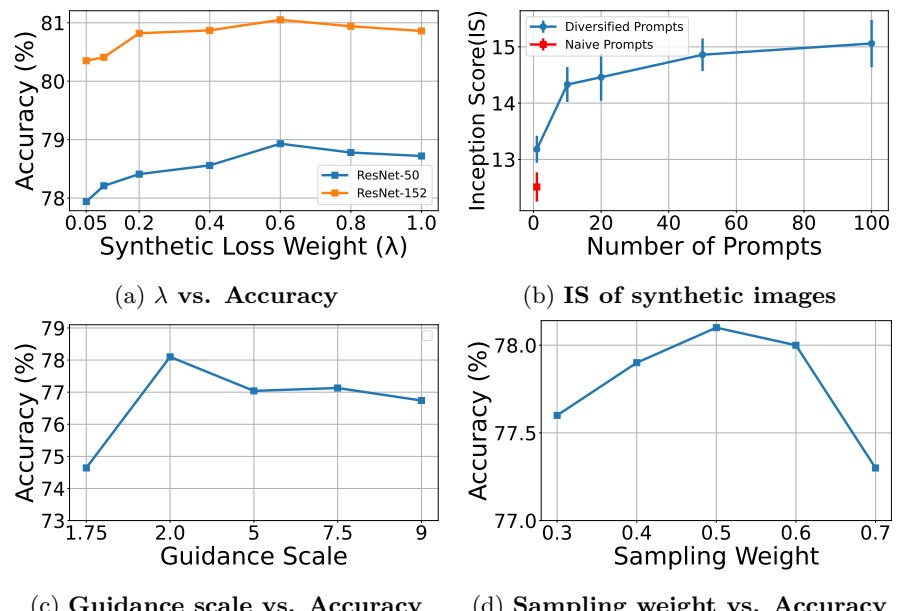

(a) $\lambda$ **vs. Accuracy**

(b) **IS of synthetic images**

(c) **Guidance scale vs. Accuracy**

(d) **Sampling weight vs. Accuracy**

Figure 7: Ablation Study: (a) **Model Sensitivity to Synthetic Loss Weights**. The model's performance remains largely unaffected by synthetic loss weights, except for very low values (e.g., 0.05). (b) **Inception Score of Synthetic Images**. Increasing the number of diversified prompts consistently enhances the Inception Score, indicating improved diversity in synthetic images. (c) **Impact of Guidance Scale**. Higher guidance scales impose stricter constraints on diffusion models following text prompts, limiting the diversity of synthetic images. Conversely, low guidance scales (e.g., 1.75) provide too much flexibility, resulting in lower quality generation. Therefore, a guidance scale of 2.0 strikes a balance between diversity and quality. (d) **Impact of sampling weight**. Higher sampling weights for synthetic data makes synthetic data more likely to be sampled at each training iteration. As shown in the figure, using a sampling weight of 0.5 (roughly balanced real and synthetic images in training) yields the best performance.

ResNet-50 and ResNet-152 is not affected much by the choice of $\lambda$ unless its value becomes very small (e.g., $\lambda < 0.2$). Therefore, we simply select $\lambda = 0.6$ for all our experiments.

**Quantifying Diversity with Inception Score**. We provide quantitative analysis of our diversification process to generate synthetic images. Specifically, we use the Inception Score (Salimans et al., 2016) as the metric to quantify diversity. We generate a total of 100 images using both naive prompt and our diversified prompts. For naive prompts, we repeatedly use the template 'a photo of a <class name>'. For diversified prompts, we adjust the number of prompts per class to evenly distribute the generation workload. As shown in Figure 7b, with the increase of different diversified prompts, the Inception Score consistently increases and is higher than that of the baseline using only naive prompt template "a photo of a <class name>". Even only using one diversified prompt from LLM, its Inception Score is still greater than that of a single prompt template demonstrating that diversified prompts not only explicitly maximize the diversity at the prompt level but also better leverage the diversity within pretrained diffusion models. This result demonstrates the efficacy of our diversification process.

**Impact of Guidance Scale**. As described in Section 3, we use a low guidance scale of 2.0 to encourage diversity in the diffusion model's generations. Figure 7c shows the recognition model's accuracy trained with

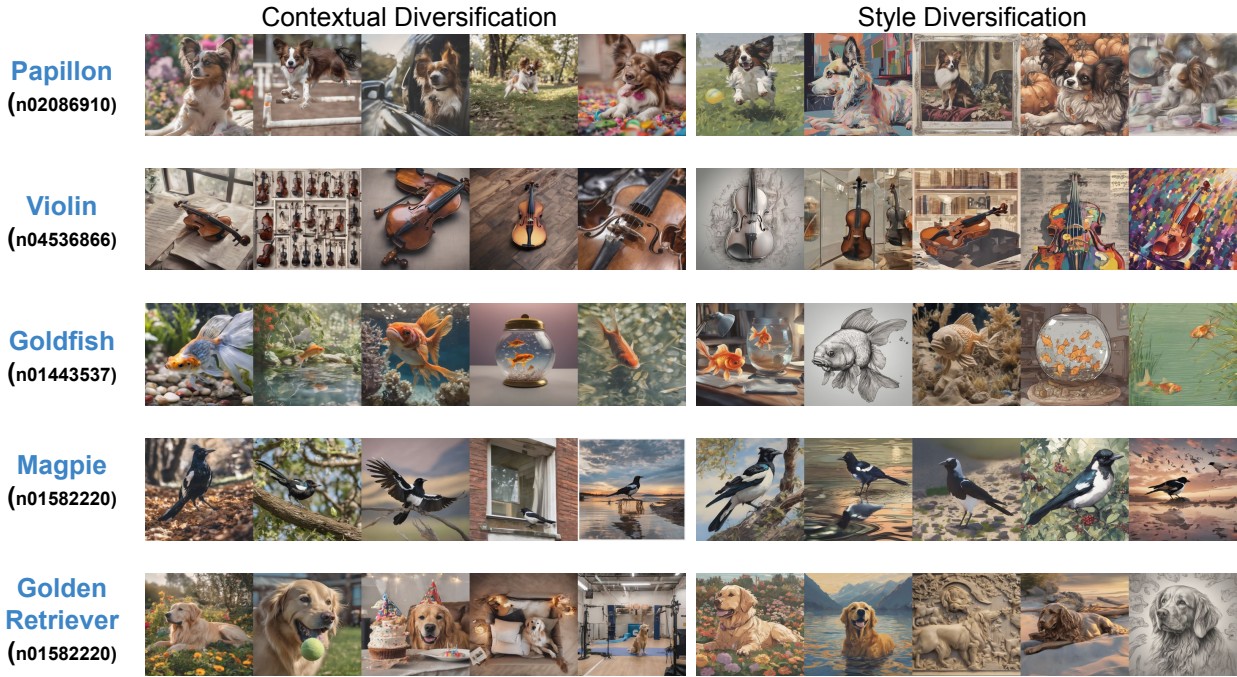

Figure 8: **Example Generations.** We show example synthetic images from our approach. Within the same class, contextual diversification creates diverse synthetic images by introducing varying contextual elements, lighting conditions, and camera angles. In contrast, style diversification alters the visual style of synthetic images. We provide example diversified prompts in Appendix for reference.

synthetic images generated with different guidance scales. The results showcase that diversified synthetic images result in the best recognition performance.

**Impact of Sampling Weight**. As described in Section 3, we use a sampling weight of 0.5 for synthetic images so that each training batch contains roughly the same number of real and synthetic images. Figure 7d shows that such choice indeed yields the best recognition model performance while increasing the sampling weight from 0.5 to 0.7 starts to hurt the final accuracy.

**Visualization of Synthetic Images.** Finally, we present example synthetic images used in our experiments in Figure 8. Our diversification approach achieves significant levels of diversification through contextual diversification (CD) and style diversification (SD). For the same class, contextual diversification introduces diversity into synthetic images by incorporating different contextual objects, varied lighting conditions and camera angles. In contrast, style diversification focuses on modifying the visual style of synthetic images.

**Limitation and Broader Impacts** We utilize off-the-shelf diffusion models to address scalability challenges in training recognition models with large-scale synthetic images. However, performance is limited by the quality of synthetic images. Future advancements in diffusion models may improve our approach. Potential negative societal impacts stem from visual recognition. Enhancing models with synthetic images, generated at minimal human costs, may lead to unintended consequences like increased surveillance.

## 5 Conclusion

We presented a scalable framework to leverage an off-the-shelf diffusion model to generate large-scale synthetic data for training visual recognition models. We demonstrated that with our methods on label ambiguity resolution, contextual and style prompt diversification, and training strategies to mitigate the real/synthetic domain gap, recognition models trained with our framework significantly outperform prior work that requires finetuning of diffusion models on target datasets. Furthermore, we also demonstrated our framework leads to strong out-of-distribution generalization performance and better ability to transfer to downstream tasks compared to previous methods. We hope our work can inspire future research on leveraging synthetic data to improve recognition models.

## Acknowledgments

This work was supported in part by NSF IIS2404180, Institute of Information & communications Technology Planning & Evaluation(IITP) grants funded by the Korea government(MSIT) (No. 2022-0-00871, Development of AI Autonomy and Knowledge Enhancement for AI Agent Collaboration) and (No. RS2022-00187238, Development of Large Korean Language Model Technology for Efficient Pre-training), and Microsoft Accelerate Foundation Models Research Program.

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

## A  Training Details

We describe the training details of our main experiments in this section (shown in Table 1). For training ResNet models, we keep most hyper-parameters the same as prior work (Azizi et al., 2023) except for batch size and learning rate. Azizi et al. (2023) uses batch size 4096 and learning rate 1.6 and we double them in our experiments to expedite training. Echoing findings in previous work (You et al., 2017), we found this difference only makes training faster without much impact on model accuracy. Also, unlike (Azizi et al., 2023) in which models are trained for 200 epochs, we train all models for 130 epochs, as we did not observe performance improvements by training them longer. Details can be found in Table 5.

| Model Parameter | ResNet-{50, 101, 152} |
|---|---|
| Epochs | 130 |
| Batch size | 8192 |
| Optimizer | Momentum |
| Learning rate | 3.2 |
| Decay method | Cosine |
| Weight decay | 1e-4 |
| Warmup epochs | 5 |
| Label smoothing | 0.1 |
| Dropout rate | 0.25 |
| Data Augment | RandAug |

Table 5: Training Details of ResNet Models.

When training large vision-transformer such as Deit-B and Deit-L (Touvron et al., 2021), we have encountered numerical stability issue[†] when using the learning rate schedule from Azizi et al (Azizi et al., 2023; Touvron et al., 2021). Therefore, we follow the refined training setup (Touvron et al., 2022) with the same number of epochs from prior work (Azizi et al., 2023; Touvron et al., 2021). The baseline model with real images trained with this new schedule achieves 81.28% top-1 accuracy for Deit-B, which is slightly lower than the baseline from prior work (Azizi et al., 2023). Note that we use the same recipe for both the real-image baselines as well as for our method. Details can be found in Table 6.

| Model | DeiT-S | DeiT-B | DeiT-L |
|---|---|---|---|
| Epochs | 300 | 300 | 300 |
| Input Size | 224 | 192 | 192 |
| Batch size | 4096 | 2048 | 2048 |
| Optimizer | AdamW | Lamb | Lamb |
| Learning rate | 0.004 | 0.003 | 0.003 |
| Learning rate decay | Cosine | Cosine | Cosine |
| Weight decay | 0.05 | 0.05 | 0.05 |
| Warmup epochs | 5 | 5 | 5 |
| Label dmoothing | 0.1 | 0.1 | 0.1 |
| Drop Path | 0.1 | 0.2 | 0.45 |
| Rand Augment | 9 | 9 | 9 |
| Mixup prob. | 0.8 | 0.8 | 0.8 |
| Cutmix prob. | 1.0 | 1.0 | 1.0 |
| Eval Crop Ratio | 0.875 | 1.0 | 1.0 |

Table 6: Training Details of Vision Transformer Models: DeiT-S Touvron et al. (2021), DeiT-B Touvron et al. (2021), and DeiT-L Touvron et al. (2021).

We use 6x synthetic data of ImageNet scale (7.2M) for ResNet models and Deit-S and 2x (2.4M) for Deit-B and Deit-L. We attribute such difference to two reasons. First, larger vision transformer models generally take longer to train per iteration. Second, those models are more sensitive to larger learning rates and linearly scaling up learning rate with batch sizes usually cause numeric stability issues. These two reasons together make training larger vision transformer models with more synthetic data take much longer than

---

[†]Such difficulty is widely observed by other community users: https://github.com/facebookresearch/deit/issues/29

|  | Training Data | $\gamma = 50$ | $\gamma = 100$ | $\gamma = 200$ |
|---|---|---|---|---|
| ResNet-50 | Real | 64.65 | 62.06 | 59.48 |
| ResNet-50 | Real + 2x Syn | **68.55** | **66.07** | **63.73** |
| ResNet-152 | Real | 67.66 | 64.91 | 62.19 |
| ResNet-152 | Real + 2x Syn | **70.81** | **68.70** | **66.31** |

Table 7: **Evaluation of Long-tail Distribution for Real Images**. We evaluate each model with imbalance ratio $\gamma$ in {50, 100, 200} by subsampling ImageNet real training images and use 2x additional synthetic data of original ImageNet scale (2.4M) from our pipeline to train our models. Compared with baseline trained with real images, our approach shows significant improvement on overall accuracy.

smaller models (more than a week in our infrastructure with 4 8-A100 GPU nodes). Therefore, we only use 2.4M synthetic data to train those larger models and significant improvement is already observed in Table 1.

## B  Low-data and Long-tail Training

As shown in Table 3 and Table 2, diversified synthetic data from our approach can also improve model training under low-data regime and long-tail distribution of real images. We provide training details of these experiments in this section. These set of experiments are evaluated with ResNet architectures.

For low-data regime, we sample 100, 200, and 500 real images per class from ImageNet real images to construct low-data training set. For models trained with our synthetic data, we keep most the training settings same as Table 5, which is used for ImageNet training except that we use 100 epochs as model usually converge faster under these conditions. For baseline models trained with real images, we use batch size 2048 and learning rate 0.88 to train these models as we found larger batch sizes produce significantly worse results for these models due to data scarcity.

The long-tailed version of ImageNet is constructed by truncating the original ImageNet training set. We use the exponential function (Cui et al., 2019) with different imbalance ratios (50, 100, and 200) to control the level of imbalance of real images. Likewise, we train all models for 100 epochs and use smaller batch sizes and learning rates as in low-data regime to train baseline models.

We define many-shot as classes with more than 800 real training images; medium-shot as classes with the number of real training images ranging from 300 to 800; and few-shot classes as classes with less than 300 training images. Results for many-shot, medium-shot, and few-shot classes are already reported in Table 3. Here, we additionally report the overall accuracy of each model in Table 7 for further reference.

## C  Combination of Diversification Methods

As discussed in Section 3, our approach involves generating synthetic data with both contextual diversification (CD) and style diversification (SD). We present the results obtained using different combinations of these diversification techniques. Specifically, we leverage 2x synthetic data of original ImageNet scale for this study.

Table 8 showcases the performance of models trained with various combinations. Using only synthetic images with CD leads to models that struggle to generalize well to datasets like ImageNet-Sketch or ImageNet-Rendition. Conversely, utilizing only images with SD results in slightly lower performance in in-domain evaluation (indicated by underlined results). Notably, combining CD and SD yields the best performance for both in-domain and out-of-domain evaluations. Therefore, we keep the same amount of synthetic data with CD and SD for all our experiments.

|  | ImNet | ImNet-V2 | Sketch | Rendition |
|---|---|---|---|---|
| 2x CD | 78.17 | 74.03 | 27.71 | 26.91 |
| 2x SD | 77.95 | 73.92 | **30.41** | **29.67** |
| 1x CD + 1x SD | **78.51** | **74.68** | 30.32 | 29.45 |

Table 8: **Results with different combinations of diversification**. We use 2x syntheitc data of original ImageNet scale and evaluate with different combinations of contextual diversification (CD) and style diversification (SD). ImageNet-val (underlined) is used for in-domain evaluation and others are for out-of-domain evaluation. Results show that using equal amount of synthetic data with CD and SD yields the best overall results.

# D    Example of Diversified Generation Prompts

Our prompt diversification procedure involves contextual diversification (CD) and style diversification (SD). To better understand our approach, we provide a list of examples of diversified prompts from CD and SD in this section.

Recall that contextual diversification prompts large language models to produce contextualized image descriptions featuring a particular object class in following four aspects: **foreground objects**, **background objects**, **lightning condition**, and **camera angle**. Descriptions of these aspects are joined together (each component is separated by comma) to form the text prompts in synthetic image generation. See below for examples:

---

**GOLDFISH (N01443537)**

*goldfish swimming in a fish tank*, *bubbles, decorative plants, pebbles*, *artificial aquarium light*, *medium shot*
*goldfish in a fishbowl with a treasure chest decoration*, *table with scattered fish food*, *room light*, *low-angle shot*
*goldfish in a backyard pond*, *surrounded by rocks, trees, garden features*, *natural sunlight*, *wide shot*
*goldfish in a school swimming together*, *fish tank with other fish, underwater plants*, *daylight*, *panoramic shot*

---

**GOLDEN RETRIEVER (N02099601)**

*golden retriever swimming in a lake*, *blue sky, mountains in the distance*, *morning*, *wide shot*
*golden retriever playing with a Frisbee*, *open field, blue sky*, *afternoon*, *long shot*
*Golden retriever wearing a service dog vest*, *busy city street*, *daylight*, *medium shot*
*golden retriever wearing a birthday hat*, *party decorations, presents*, *indoor lightning*, *overhead shot*

---

**LEOPARD (N02128385)**

*resting leopard, tail curled*, *dense jungle foliage, trees,*, *dappled sunlight*, *medium shot*
*leopard crouching in hunting position*, *dense jungle, trees and foliage*, *golden hour*, *close-up shot*
*leopard drinking from a river*, *reflecting water, rocky terrain, distant mountains*, *harsh daylight*, *wide shot*
*leopard grooming itself*, *fallen tree trunk, colorful bird perched nearby*, *late afternoon sunlight*, *macro shot*

---

**PRETZEL (N07695742)**

*pretzel held in hand*, *a cup of coffee, a newspaper,*, *natural daylight*, *close-up shot*
*pretzel with a drizzle of chocolate sauce*, *a plate of assorted desserts*, *dimmed lightning*, *medium shot*
*pretzel in a bakery display case*, *freshly baked bread, pastries*, *soft spotlight*, *close-up shot*
*pretzel being served with a side of sausages*, *traditional German beer garden*, *outdoor daylight*, *wide shot*

---

The prompts with contextual diversification are then combined with the prefix "*A photograph of*" to generate photo-realistic synthetic images. On the other hand, the style diversification replaces *photograph* with each

| Training Data | Flowers | Pets | Food | DTD |
|---|---|---|---|---|
| KNN | | | | |
| Real | 66.75 | 88.06 | 52.25 | 57.18 |
| Real + 6x Synthetic | **67.26** | **90.27** | **55.44** | **61.70** |
| Linear Probing | | | | |
| Real | 82.12 | 90.18 | 67.18 | 64.89 |
| Real + 6x Synthetic | **82.46** | **92.26** | **68.81** | **66.48** |

Table 9: **Transfer Learning Evaluations**. We compare the transfer learning performance of a ResNet-50 model trained with only real ImageNet images, to a model trained with real + 6x synthetic ImageNet images. Results show that models trained with synthetic data (ours) demonstrate stronger capability to transfer to other tasks.

of the style keyword that is crawled from LLMs to form style diversified prompts. The list of style keywords is shown below:

> STYLES FROM LLM
>
> *Sketch, Painting, Illustration, Digital rendering, Print, Comic-style depiction, Manga-style depiction, Pixel art representation, Tattoo design, Graffiti-style portrayal, Watercolor, Oil painting, Charcoal drawing, Pastel drawing, Stencil art, Collage, Mosaic, Silhouette, Pop art version, Sculpture, Origami, Embroidery, Quilt pattern, Stained glass design, Woodcut, Etching, Lithograph, Screen print, Relief carving, Bronze casting, Glass blowing, Ceramic pottery, Tapestry, Fresco, Mural, Doodle, Cartoon, Animation, 3D model, Wireframe model, CGI, Virtual reality model, Augmented reality model, Hologram, Gouache painting, Ink wash painting, Digital painting, Stencil graffiti, Airbrush art, Pointillism, Impasto painting, Engraving, Linocut, Marquetry, Papercut, Batik design, Cross-stitch pattern, Macramé design, Beadwork design, Sand sculpture*

## E   Synthetic Image Generation Details

We provide details of synthetic data generation in addition to the implementation details described in Section 4. Specifically, we set the guidance scale to 2.0 and use 50 sampling steps to generate synthetic images. Synthetic images are sampled at resolution of 1024x1024 and downsampled to 256x256 for storage. For faster generation, fp16 inference is used for all synthetic images.

## F   Transfer Learning for Other Classification Tasks

To examine the quality of representation learned from our synthetic data, we further evaluate the transfer learning performance of our models on various classification tasks. Specifically, we evaluate on four datasets: Oxford Flowers (Nilsback & Zisserman, 2008), Oxford Pets (Parkhi et al., 2012), Food-101 (Bossard et al., 2014), and Describable Textures (Cimpoi et al., 2014). Following the standard transfer learning evaluation protocol, we use both KNN and linear probing to evaluate transfer learning performance.

As shown in Table 9, ResNet-50 models trained with diversified synthetic data demonstrate stronger performance in transferring to other classification tasks under both k-nearest neighbor and linear probing evaluation. On average our model achieves +2.61% and +1.41% accuracy improvement with KNN and linear probing respectively.

