# OpenReview forum: "Diversify, Don't Fine-Tune: Scaling Up Visual Recognition Training with Synthetic Images"
_TMLR — Accepted by TMLR_

### Review · Reviewer_JQcx · 2024-09-30

**Summary Of Contributions:**

In this paper, the authors propose method to improve models by training them using off-the-shelf generative models. They set up an automated pipeline to generate labelled images while identifying two issues with using generative models -- lack of diversity in generated images and domain shift. To solve these they propose diversification techniques for both context and style, and use auxiliary batch-norm to mitigate domain shifts. They show that using their method, models can improve their performance by training on a much larger proportion of generated images without performance collapse.

**Audience:**

Yes

**Claims And Evidence:**

Yes

**Requested Changes:**

Only the ones listed under "minor clarity issues" in weaknesses above.

**Strengths And Weaknesses:**

### Strengths

- The paper is well motivated and timely -- using generated data to train models has the potential to reduce the need for manually annotated data while giving us better models
- The paper has very clearly defined and targeted insights
- The method is conceptually intuitive and clearly explained
- The authors show good empirical results with their method, and their evaluation is reasonably rigorous.

### Weaknesses

- In the comparison with prior work (Azizi et al. 2023): it would be interesting to see how much difference just adding auxiliary batch norm parameters would make to their method.

Minor clarity issues:
- It's not clear where/how LAR output is used later (Fig. 2)
- The description of LAR could use a figure (flowchart) to explain the steps exactly.

---

### Review · Reviewer_uXdr · 2024-10-12

**Summary Of Contributions:**

This work proposes a well-designed pipeline for generating diverse synthetic data to train image classifiers. The main technical contribution of the work lies in (1) solving the label ambiguity through curating concept embeddings with the vision-language model, (2) prompt engineering to diversify generation in terms of camera angle, illumination, style, etc., (3) regularization for real-synthetic data domain gap in the loss function. The general generation framework in this work is in line with existing literature, while the proposed pipeline has been empirically proven effective with state-of-the-art performance achieved.

**Audience:**

Yes

**Broader Impact Concerns:**

I don't have any prominent ethical concern

**Claims And Evidence:**

Yes

**Requested Changes:**

Please see the weakness section.

**Strengths And Weaknesses:**

## Strength:
1. The paper is well-written and easy to follow
2. The empirical result in ImageNet-val looks promising and outperforms [1].
3. The design choice of specific hyperparameters is well-justified with the corresponding ablation study.

## Weakness:
1. In the experiment, having a baseline where the image is generated with the same backbone model but without the proposed module would be helpful. The current baseline [1] uses a different backbone model, Imagen, which is not open-sourced, thus making the improvement margin less informative to reflect the true benefit of the proposed methods.
2. For a similar reason as in the above, the experiment on O.O.D datasets has no baseline but only real data. I assume this is because there are no corresponding O.O.D dataset results reported in the original paper [1].
3. The paper says, "... we use the latest variation of LDM ..." it would be good if the authors explicitly say which model is used, as the definition of "latest" may change.

## Summary
Overall, I think this is valuable work, with good motivation for the method and a solid ablation study on the practical design. I believe this paper can be interesting to researchers interested in synthetic training data generation.

[1] Azizi, S., Kornblith, S., Saharia, C., Norouzi, M. and Fleet, D.J., 2023. Synthetic data from diffusion models improves imagenet classification. TMLR.

---

### Review · Reviewer_sBLh · 2024-10-14

**Summary Of Contributions:**

The work presents an improved synthetic data generation pipeline for large scale classification (ImageNet). Specifically they,
- Improve classification performance by +1.10% on ResNet-50 compared to previous work (Azizi et al., 2023). Improvements are also seen for larger ResNet's and DeiT models.
- Eliminate the need for fine-tuning text-to-image generators on the target dataset, an important simplification compared to prior work (Azizi et al., 2023). They do this by resolving the label ambiguity problem (e.g. "crane" the bird vs "crane" the machine) associated with pre-trained generators by using LLMs and CLIP to generate a short un-ambiguous description of the class (e.g. "crane, a long legged bird"), used during generation.
- Qualitatively and quantitatively improve the diversity of generated images by using LLMs to yield diverse generation prompts.
- Significant classification performance improvements are also seen in out-of-domain tasks such as ImageNet-V2, ImageNet-Sketch and ImageNet-Rendition.

**Audience:**

Yes

**Broader Impact Concerns:**

No concerns.

**Claims And Evidence:**

No

**Requested Changes:**

I agree that the method is valuable, as the simplification over prior approaches and increase in diversity of the generated images is significant.

However, the first two weaknesses of the work are critical to the validity of the results and the scientific knowledge gained from the paper, so I cannot recommend acceptance unless the following changes are made:
- **Ensure there is no train-test contamination**: Specify the exact text-to-image generative model used and the dataset it was trained on, and ensure there is no train-test contamination. i.e. make sure the ImageNet validation set is not part of the generator's training set.
- **Reword the claims regarding synthetic data outnumbering real data**: Since real data is artificially up-weighted to 50%, trained models see equal amounts of real and synthetic data during training. Hence the claims made regarding "consistent accuracy improvement as the number of synthetic samples increases even after they outnumber the real samples" is not fully precise. For clarity, I am absolutely in favor of artificially up-sampling real data, and in no way does it diminish the results of the work. My objection is only in the language used. In order to justify the original claims, the authors would need to show similar results without the artificial up-sampling.
- **Apples-to-apples comparison without artificial up-weighting in Figure 3**: This artificial up-weighting is not done in (Azizi et al., 2023), so the comparison in Figure 3 is not apples-to-apples. I think an additional experiment where artificial up-weighting of real data is not used needs to be added. The main focus of the work are insights regarding synthetic data scaling, so this experiment is critical for the scientific knowledge contributed by the paper.

**Strengths And Weaknesses:**

Strengths
- Eliminating the need for fine-tuning the generative model is a significant simplification that is very useful for end-users.
- The diversity of the generated images is qualitatively and quantitatively stronger than that of prior work.

Weaknesses
- The work does not specify the exact text-to-image generative model used, specifically the dataset it was trained on. This is of critical importance since ensuring there is no train-test contamination is crucial to the validity of the results.
- While the amount of generated data is scaled up to 12x that of real data, the authors weigh the sampling probabilities such that the model sees 50% real data and 50% synthetic data during training. Hence, synthetic data never outnumbers the amount of real samples during training. This makes the results of Figure 3 and section 4.1 potentially misleading or overclaiming. Additionally, the comparison to (Azizi et al., 2023) in Figure 3 is not apples-to-apples, as that work did not artificially up-weight real data.
- The potency of the long-tail distribution experiments is greatly lessened by the fact that the generative model has seen many examples of the artificially un-likely classes. In order to have rigorous experiments, the un-likely classes should also be un-likely for the generative model.

---

### Decision · Action_Editor_XX4K · 2024-12-06

**Recommendation:** Accept as is

**Comment:**

This paper presents an improved synthetic data generation pipeline for training large scale image classification. The core contributions lie in
1) eliminating the need for fine-tuning, significantly simplifying the process and improving speed and
2) disambiguate labels (e.g., crane), and
3) diversification (e.g., foreground/background objects, lighting, viewpoint, styles).

All three reviewers agree that the paper showcases solid experimental results validating the effectiveness of the proposed synthetic data generation pipeline. Compared to prior method that relies on fine-tuning (Azizi et al., 2023), this approach achieves improved results without fine-tuning will be useful for end-users.

**Audience:**

Yes, the paper will be of interests to ML community.

**Claims And Evidence:**

Yes, the claims are supported by convincing experimental evidence.